# Adaptation to the cervical environment is associated with increased antibiotic susceptibility in *Neisseria gonorrhoeae*

Kevin C. Ma [1,7], Tatum D. Mortimer [1,7], Allison L. Hicks [1], Nicole E. Wheeler [2],
Leonor Sánchez-Busó [2], Daniel Golparian [3], George Taiaroa[4], Daniel H. F. Rubin[1], Yi Wang[1],
Deborah A. Williamson [4], Magnus Unemo[3], Simon R. Harris[5] & Yonatan H. Grad [1,6 ✉]

*Neisseria gonorrhoeae* is an urgent public health threat due to rapidly increasing incidence and antibiotic resistance. In contrast with the trend of increasing resistance, clinical isolates that have reverted to susceptibility regularly appear, prompting questions about which pressures compete with antibiotics to shape gonococcal evolution. Here, we used genome-wide association to identify loss-of-function (LOF) mutations in the efflux pump *mtrCDE* operon as a mechanism of increased antibiotic susceptibility and demonstrate that these mutations are overrepresented in cervical relative to urethral isolates. This enrichment holds true for LOF mutations in another efflux pump, *farAB*, and in urogenitally-adapted versus typical *N. meningitidis*, providing evidence for a model in which expression of these pumps in the female urogenital tract incurs a fitness cost for pathogenic *Neisseria*. Overall, our findings highlight the impact of integrating microbial population genomics with host metadata and demonstrate how host environmental pressures can lead to increased antibiotic susceptibility.

[1] Department of Immunology and Infectious Diseases, Harvard T.H. Chan School of Public Health, Boston, MA, USA. [2] Centre for Genomic Pathogen Surveillance, Wellcome Sanger Institute, Wellcome Genome Campus, Hinxton, Cambridgeshire, UK. [3] WHO Collaborating Centre for Gonorrhoea and other STIs, Swedish Reference Laboratory for STIs, Faculty of Medicine and Health, Örebro University, Örebro, Sweden. [4] Microbiological Diagnostic Unit Public Health Laboratory, Department of Microbiology and Immunology, The University of Melbourne at The Peter Doherty Institute for Infection and Immunity, Melbourne, VIC, Australia. [5] Microbiotica Ltd, Biodata Innovation Centre, Wellcome Genome Campus, Hinxton, Cambridgeshire, UK. [6] Division of Infectious Diseases, Brigham and Women's Hospital and Harvard Medical School, Boston, MA, USA. [7] These authors contributed equally: Kevin C. Ma, Tatum D. Mortimer. ✉email: ygrad@hsph.harvard.edu

**N**eisseria gonorrhoeae is the causative agent of the sexually transmitted disease gonorrhea. Antibiotics have played a key role in shaping gonococcal evolution[1–3], with *N. gonorrhoeae* gaining resistance to each of the first-line antibiotics used to treat it[4–6]. As *N. gonorrhoeae* is an obligate human pathogen, the mucosal niches it infects—most commonly including the urethra, cervix, pharynx, and rectum—must also influence its evolution[7]. The gonococcal phylogeny suggests the interaction of these factors, with an ancestral split between a drug-susceptible lineage circulating primarily in heterosexuals and a drug-resistant lineage circulating primarily in men who have sex with men[3].

Despite the deeply concerning increase in antibiotic resistance reported in gonococcal populations globally[8], some clinical isolates of *N. gonorrhoeae* have become more susceptible to antibiotics[9,10]. This unexpected phenomenon prompts questions about which environmental pressures could be drivers of increased susceptibility and the mechanisms by which suppression or reversion of resistance may occur. To address these questions, we analyzed the genomes of a global collection of clinical isolates together with patient demographic and clinical data to identify mutations associated with increased susceptibility and define the environments in which they appear. We find that loss-of-function mutations in the efflux pump component *mtrC* are significantly associated with both antibiotic susceptibility and cervical infections, demonstrating how antibiotic and mucosal niche selective pressures intersect. In support of a model in which efflux pump expression incurs a cost in this niche, we also observe enrichment of loss-of-function mutations in cervical isolates in another efflux pump in *N. gonorrhoeae* and in urogenitally-adapted *N. meningitidis*. Our findings demonstrate how shifts in environmental pressures experienced by pathogenic *Neisseria* can lead to loss of efflux pump function and suppression of antibiotic resistance.

## Results

**Unknown genetic loci influence antibiotic susceptibility.** We first assessed how well variation in antibiotic resistance phenotype was captured by the presence and absence of known resistance markers. To do so, we assembled and examined a global dataset comprising the genomes and minimum inhibitory concentrations (MICs) of 4852 isolates collected across 65 countries and 38 years (Fig. 1, Supplementary Table 1). We modeled log-transformed MICs using multiple regression on a panel of experimentally characterized resistance markers for the three most clinically relevant antibiotics[5,11,12] (Supplementary Data 1). This enabled us to make quantitative predictions of MIC based on known genotypic markers and to assess how well these markers predicted true MIC values. For the macrolide azithromycin, we observed that 434/4505 (9.63%) isolates had predicted MICs that deviated by two dilutions or more from their reported phenotypic values. The majority (59.4%) of these isolates had MICs that were lower than expected, indicative of increased susceptibility unexplained by the genetic determinants in our model. Overall MIC variance explained by known resistance mutations was relatively low (adjusted $R^2 = 0.667$), in agreement with prior studies that employed whole-genome supervised learning algorithms to predict azithromycin resistance[13]. MIC variance explained by known resistance mutations was also low for ceftriaxone (adjusted $R^2 = 0.674$) but higher for ciprofloxacin (adjusted $R^2 = 0.937$), with 2.02% and 2.90% of strains, respectively, exhibiting two dilutions or lower reported MICs compared to predictions, similarly indicating unexplained susceptibility. The predictive modeling results, therefore, suggested unknown modifiers that promote susceptibility for multiple classes of antibiotics in *N. gonorrhoeae*.

**GWAS identifies a susceptibility-associated variant in *mtrC*.** To identify novel antibiotic susceptibility loci in an unbiased manner, we conducted a bacterial genome-wide association study (GWAS). We used a linear mixed model framework to control for population structure, and we used unitigs constructed from genome assemblies to capture SNPs, indels, and accessory genome elements (see "Methods")[14–16]. Unitigs are a flexible representation of the genetic variation across a dataset that are constructed using compacted de Bruijn graphs and have been previously applied as markers for microbial GWAS[16]. We performed a GWAS on the sequences of 4505 isolates with associated azithromycin MICs using a Bonferroni-corrected significance threshold of $3.38 \times 10^{-7}$. The linear mixed model adequately controlled for population structure (Supplementary Fig. 1), and the proportion of phenotypic MIC variance attributable to genotype (i.e., narrow-sense heritability) estimated by the linear mixed model was high ($h^2 = 0.97$). In line with this, we observed highly significant unitigs with positive effect sizes corresponding to the known resistance substitutions C2611T and A2059G (*E. coli* numbering) in the 23S ribosomal RNA gene (Fig. 2)[17]. The next most significant variant was a unitig associated with increased susceptibility that mapped to *mtrC* (β, or effect size on the $\log_2$-transformed MIC scale $= -2.82$, 95% CI $[-3.06, -2.57]$; $p$-value $= 2.81 \times 10^{-108}$). Overexpression of the *mtrCDE* efflux pump operon has been shown to decrease gonococcal susceptibility to a range of hydrophobic acids and antimicrobial agents[4,18], and conversely, knockout of the pump results in multidrug hypersusceptibility[19]. To assess whether this *mtrC* variant was associated with increased susceptibility to other antibiotics, we performed GWAS for ceftriaxone (for which MICs were available from 4497 isolates) and for ciprofloxacin (4135 isolates). We recovered known ceftriaxone resistance mutations including recombination in the *penA* gene and ciprofloxacin resistance substitutions in DNA gyrase (*gyrA*). In agreement with the known pleiotropic effect of the MtrCDE efflux pump[19], we observed the same *mtrC* unitig at genome-wide significance associated with increased susceptibility to both ceftriaxone (β $= -1.18$, 95% CI $[-1.34, -1.02]$; $p$-value $= 2.00 \times 10^{-44}$) and ciprofloxacin (β $= -1.29$, 95% CI $[-1.54, -1.04]$; $p$-value $= 1.87 \times 10^{-23}$) (Fig. 2). Across all three drugs, heritability estimates for this *mtrC* variant were comparable to that of prevalent major resistance determinants (azithromycin $h^2$: 0.323; ceftriaxone $h^2$: 0.208; ciprofloxacin $h^2$: 0.155), indicating that unexplained susceptibility in our model could be partially addressed by the inclusion of this mutation.

Annotation of the *mtrC* unitig revealed that it represented a two base pair deletion in a 'GC' dinucleotide hexarepeat, leading to early termination of *mtrC* translation and probable loss of MtrCDE activity[20] (Fig. 2 inset). We also checked whether the two base pair deletion would affect recognition by any of the gonococcal methylases[21], but no methylase target motif sites mapped to the hexarepeat or its direct surrounding sequences. A laboratory-generated gonococcal mutant with a four base pair deletion in this same *mtrC* dinucleotide hexarepeat exhibited multi-drug susceptibility[20], and clinical gonococcal isolates hypersensitive to erythromycin were shown to have mutations mapping to this locus[22]. To directly test the hypothesis that the two base pair deletion also contributed to increased susceptibility for the panel of antibiotics we examined, we complemented the mutation in a clinical isolate belonging to the multidrug-resistant lineage ST-1901[23] and observed significant increases in MICs for all three antibiotics, as predicted by the GWAS (Supplementary Table 2).

We searched for additional *mtrC* loss-of-function (LOF) mutations and found six clinical isolates with genomes encoding indels outside of the dinucleotide hexarepeat that also were

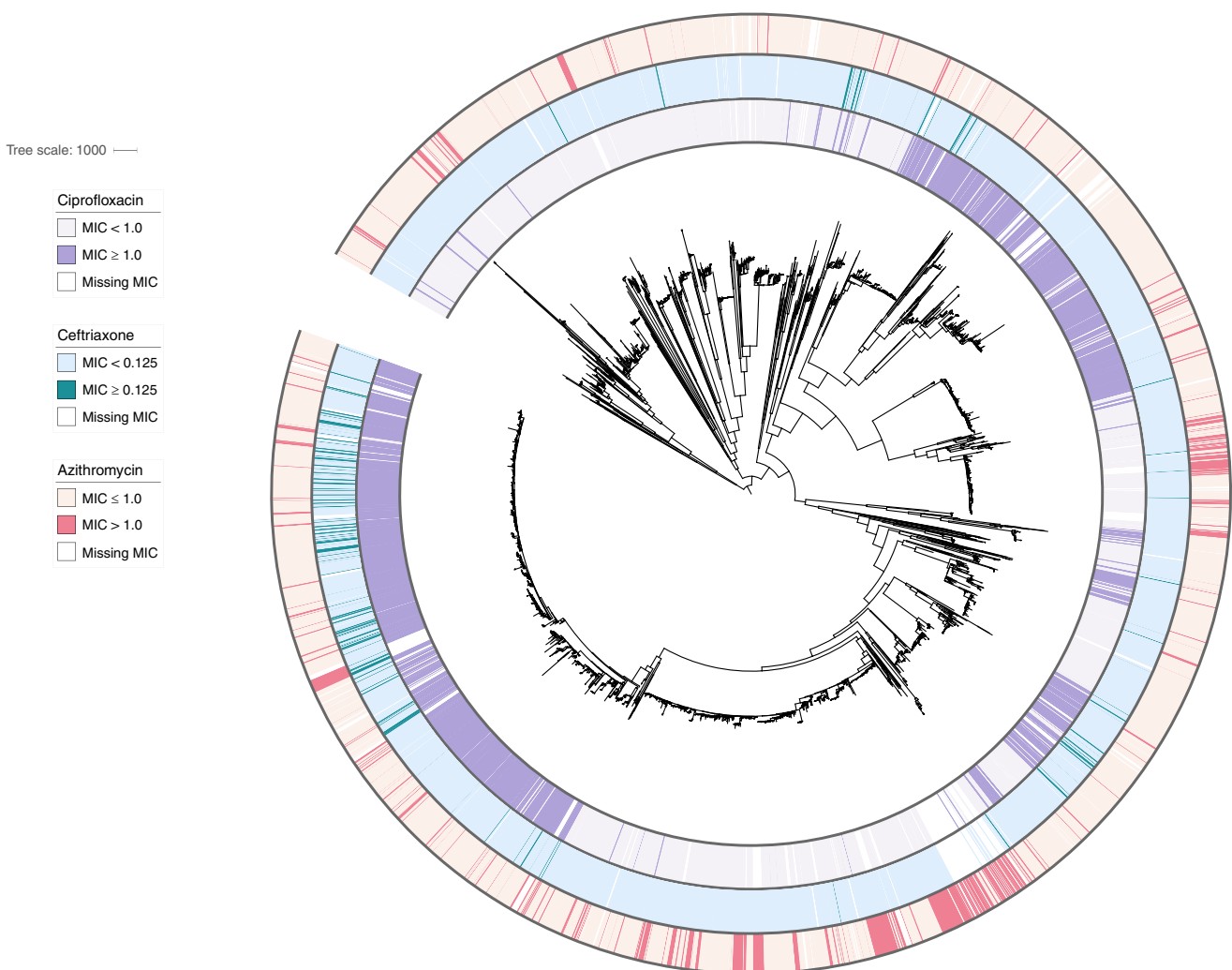

**Fig. 1 Population structure and susceptibility profile of *N. gonorrhoeae* global meta-analysis collection.** A midpoint rooted recombination-corrected maximum likelihood phylogeny of 4852 genomes based on 68697 SNPs (Supplementary Table 1) was annotated with binarized resistance (ciprofloxacin) or decreased susceptibility (azithromycin, ceftriaxone) values. Annotation rings are listed in order of ciprofloxacin, ceftriaxone, and azithromycin from innermost to outermost. For ciprofloxacin, MIC < 1 μg/ml is light purple, and MIC ≥ 1 μg/ml is dark purple. For ceftriaxone, MIC < 0.125 μg/ml is light blue, and MIC ≥ 0.125 μg/ml is dark blue. For azithromycin, MIC ≤ 1 μg/ml is light pink, and MIC > 1 μg/ml is dark pink. Branch length represents total number of substitutions after removal of predicted recombination.

associated with increased susceptibility (Supplementary Fig. 2A). Ten isolates that had acquired the two base pair deletion also have a two base pair insertion elsewhere in *mtrC* that restores the original coding frame, suggesting that loss of MtrC function may be reverted by further mutation or recombination (Supplementary Fig. 2A). In line with this, *mtrC* LOF mutations have emerged numerous times throughout the phylogeny (Supplementary Fig. 3), indicative of possible repeated losses of a dinucleotide in the hexarepeat region due to DNA polymerase slippage, which may occur at a higher rate than single nucleotide nonsense mutations[24]. In total, including all strains with *mtrC* frameshift mutations and excluding revertants, we identified 185 isolates (3.82%) that encoded a LOF allele of *mtrC* (Supplementary Table 3). Presence of the *mtrC* LOF mutation in isolates with known resistance markers was correlated with significantly reduced MICs (Supplementary Fig. 4), and inclusion of *mtrC* LOF mutations in our linear model increased adjusted $R^2$ values (azithromycin: 0.667–0.704; ceftriaxone: 0.674–0.690; ciprofloxacin: 0.937–0.939), decreased the proportion of strains with unexplained susceptibility (azithromycin: 5.73%–3.88%;

ceftriaxone: 2.02%–1.73%; ciprofloxacin: 2.90%–2.42%), and significantly improved model fit (*p*-value < 2.2 × 10$^{-16}$ for all three antibiotics; Likelihood-ratio $\chi^2$ test for nested models). *mtrC* LOF strains were identified in 28 of the 66 countries surveyed and ranged in isolation date from 2002 to 2017. Because most strains in this dataset were collected within the last two decades, we also examined a dataset of strains collected in Denmark from 1928 to 2013 to understand the historical prevalence of *mtrC* LOF mutations[25]. We observed an additional 10 strains with the 'GC' two base pair deletion ranging in isolation date from 1951 to 2000, indicating that *mtrC* LOF strains have either repeatedly arisen or persistently circulated for decades. Our results demonstrate that a relatively common mechanism of gonococcal acquired antibiotic susceptibility is a two base pair deletion in *mtrC* and that such mutations are globally and temporally widespread.

**Loss of MtrCDE pump is associated with cervical infection.** The MtrCDE pump has been demonstrated to play a critical role in gonococcal survival in the presence of human neutrophils and in

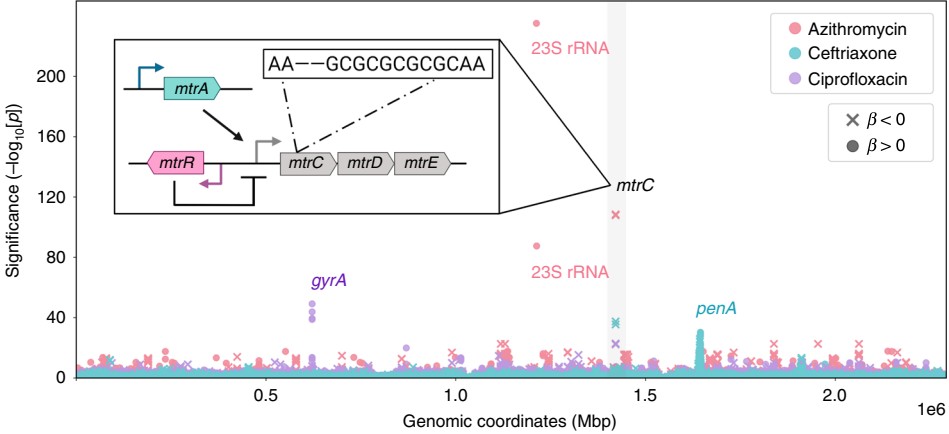

**Fig. 2 GWAS identifies a variant mapping to *mtrC* associated with increased susceptibility.** The Manhattan plot shows negative log$_{10}$-transformed *p*-values (calculated using likelihood-ratio tests in the GWAS) for the association of unitigs with MICs to azithromycin (pink, *n* = 4505), ceftriaxone (blue, *n* = 4497), and ciprofloxacin (purple, *n* = 4135). The sign of the GWAS regression coefficient β (with positive indicating an association with increased resistance and negative indicating an association with increased susceptibility) is indicated by an X for β < 0 and a dot for β > 0. Labels indicate known influential resistance determinants, and the *mtrC* variant associated with increased susceptibility was highlighted in gray. A full list of the annotated significant unitigs for each antibiotic can be found in Supplementary Data 2. Inset: schematic of the *mtr* genetic regulon including structural genes *mtrCDE*, the activator *mtrA*, and the repressor *mtrR*. The approximate genomic location within *mtrC* and specific nucleotide change of the *mtrC* GWAS variant relative to the gonococcal NCCP11945 reference genome (i.e., a two base pair deletion in a 'GC' dinucleotide repeat) is shown.

the female murine genital tract model of gonococcal infection, and overexpression of *mtrCDE* results in substantial fitness benefits for dealing with both antimicrobial and environmental pressures[26–29]. The relative frequency of the *mtrC* LOF mutations we observe (occurring in ~1 in every 25 isolates) thus seems unusual for a mutation predicted to be deleterious for human infection. *mtrC* LOF strains do not grow more or less quickly in vitro than *mtrC* wild-type strains, indicating that this mutation does not confer a simple fitness benefit due to reduced energetic cost[22,26,30]. Instead, we hypothesized that there are unique environments that select for non-functional efflux pump.

We aggregated patient-level metadata across included studies on sex partner preferences and anatomical site of infection. Sexual behavior and *mtrC* genotypic information was available for 1975 isolates from individual patients. There was a significant association between *mtrC* LOF and sexual behavior (*p*-value = 0.04021; two-sided Fisher's exact test) (Fig. 3a, Supplementary Table 4), and *mtrC* LOF occurred more often in isolates from men who have sex with women (MSW) (28/626, 4.47%) compared to isolates from men who have sex with men (MSM) (31/1189, 2.61%) (OR = 1.75, 95% CI [1.00–3.04], *p*-value = 0.037; two-sided Fisher's exact test). To understand whether anatomical selective pressures contributed to this enrichment, we analyzed the site of infection and *mtrC* genotypic information available for 2730 isolates. *mtrC* LOF mutations were significantly associated with site of infection (*p*-value = 6.49 × 10$^{-5}$; two-sided Fisher's exact test) and were overrepresented particularly in cervical isolates: 16 out of 129 (12.4%) cervical isolates contained an *mtrC* LOF mutation compared to 82 out of 2249 urethral isolates (3.65%; OR = 3.74, 95% CI [1.98–6.70], *p*-value = 4.71 × 10$^{-5}$; two-sided Fisher's exact test), 3 out of 106 pharyngeal isolates (2.83%; OR = 4.83, 95% CI [1.33–26.63], *p*-value = 0.00769; two-sided Fisher's exact test), and 4 out of 246 rectal isolates (1.63%; OR = 8.52, 95% CI [2.67–35.787], *p*-value = 2.39 × 10$^{-5}$; two-sided Fisher's exact test) (Fig. 3b, Supplementary Table 5). Because our meta-analysis collection comprised datasets potentially biased by preferential sampling for drug-resistant strains, we validated our epidemiological associations on a set of 2186 sequenced isolates, corresponding to all cultured isolates of *N. gonorrhoeae* in the state of Victoria, Australia in 2017[31]. We

again observed significant associations between *mtrC* LOF and sexual behavior (*p*-value = 0.0180; two-sided Fisher's exact test) as well as the anatomical site of infection (*p*-value = 0.0256; two-sided Fisher's exact test) (Supplementary Fig. 5, Supplementary Tables 6 and 7). *mtrC* LOF mutations were again overrepresented in cervical isolates: 9 out of 227 (3.96%) cervical isolates contained an *mtrC* LOF mutation compared to 15 out of 882 urethral isolates (1.70%; OR = 2.38, 95% CI [0.91–5.91], *p*-value = 0.0679; two-sided Fisher's exact test), 3 out of 386 pharyngeal isolates (0.78%; OR = 5.26, 95% CI [1.29–30.51], *p*-value = 0.0117; two-sided Fisher's exact test), and 7 out of 632 rectal isolates (1.11%; OR = 3.68, 95% CI [1.20–11.78], *p*-value = 0.0173; two-sided Fisher's exact test). These results indicate that environmental pressures unique to female urogenital infection may select for loss of the primary gonococcal efflux pump resulting in broadly increased susceptibility to antibiotics and host-derived antimicrobial peptides.

**mtrA LOF offers an additional level of adaptive regulation.** The association of *mtrC* LOF mutations with cervical specimens suggests that other mutations that downregulate expression of the *mtrCDE* operon should also promote adaptation to the cervical niche. The MtrCDE efflux pump regulon comprises the MtrR repressor and the MtrA activator (Fig. 2 inset), the latter of which exists in two allelic forms: a wild-type functional gene capable of inducing *mtrCDE* expression and a variant rendered non-functional by an 11-bp deletion near the 5' end of the gene[32] (Supplementary Fig. 2B). Knocking out *mtrA* has a detrimental effect on fitness in the gonococcal mouse model, and epistatic *mtrR* mutations resulting in overexpression of *mtrCDE* compensate for this fitness defect by masking the effect of the *mtrA* knockout[27]. Prior work assessing the genomic diversity of *mtrA* in a set of 922 primarily male urethral specimens found only four isolates with the 11-bp deletion (0.43%)[33], seemingly in agreement with the in vivo importance of *mtrA*. However, in our global meta-analysis dataset, 362/4842 isolates (7.48%) were predicted to be *mtrA* LOF, of which the majority (357/362, 98.6%) were due to the 11-bp deletion. Of the 4842 isolates, 268 (5.53%) had *mtrA* LOF mutations in non-*mtrCDE* overexpression backgrounds (as

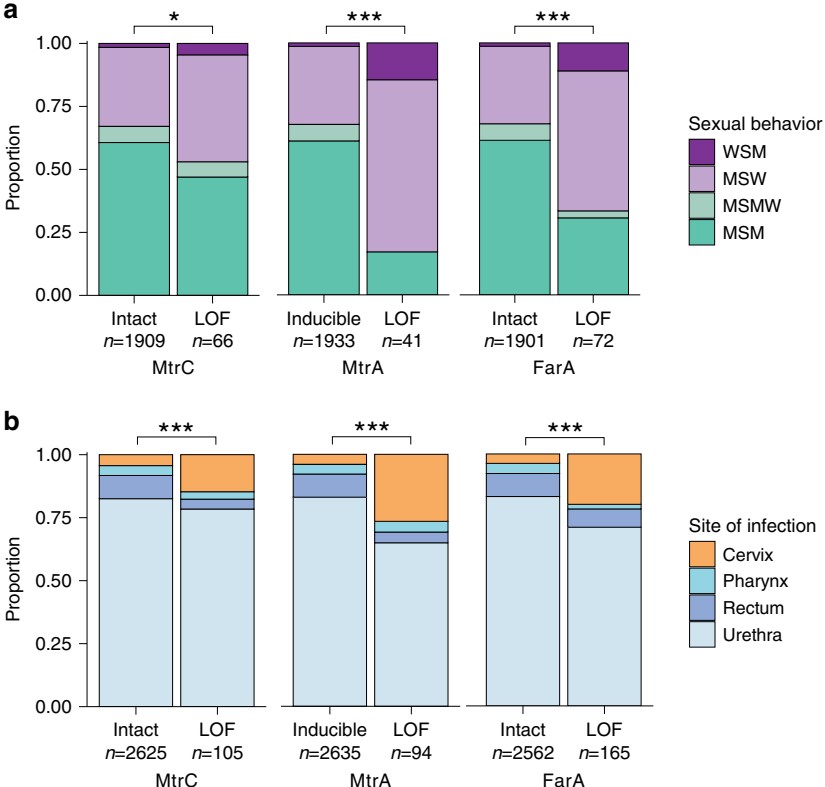

**Fig. 3 Gonococcal *mtrC*, *mtrA*, and *farA* LOF mutations are associated with cervical infection. a** Sexual behavior of patients infected with isolates with either intact or LOF alleles of *mtrC* (left), *mtrA* (middle), or *farA* (right). **b** Site of infection in patients infected with isolates with either intact or LOF alleles of *mtrC* (left), *mtrA* (middle), or *farA* (right). *mtrA* alleles were predicted as LOF only in the absence of other epistatic Mtr overexpression mutations. Statistical significance between genotype and patient metadata was assessed by two-sided Fisher's exact test: *$p < 0.05$, **$p < 0.01$, and ***$p < 0.001$. Exact $p$-values from left to right for analyses in **a** were 0.04021, $1.81 \times 10^{-11}$, $5.06 \times 10^{-10}$ and for **b** were $6.49 \times 10^{-5}$, $1.64 \times 10^{-12}$, $1.78 \times 10^{-12}$. WSM = women who have sex with men, MSW = men who have sex with women, MSMW = men who have sex with men and women, MSM = men who have sex with men.

defined by the absence of known *mtrR* promoter or coding sequence mutations or *mtrCDE* mosaic alleles) and therefore not epistatically masked (Supplementary Table 8). We repeated our epidemiological associations on these *mtrA* LOF strains without concurrent overexpression mutations and observed highly significant associations with reported patient sexual behavior ($p$-value = $1.81 \times 10^{-11}$; two-sided Fisher's exact test) and site of infection ($p$-value = $1.64 \times 10^{-12}$; two-sided Fisher's exact test) (Fig. 3, Supplementary Tables 9–10). As with *mtrC* LOF mutations, *mtrA* LOF mutations were significantly overrepresented in cervical isolates: 25 out of 129 (19.4%) cervical isolates contained an *mtrA* LOF mutation compared to 61 out of 2248 urethral isolates (2.71%; OR = 8.60, 95% CI [4.96–14.57], $p$-value = $4.60 \times 10^{-13}$; two-sided Fisher's exact test), 4 out of 106 pharyngeal isolates (3.78%; OR = 6.09, 95% CI [2.00–24.93], $p$-value = 0.000240; two-sided Fisher's exact test), and 4 out of 246 rectal isolates (1.63%; OR = 14.43, 95% CI [4.81–58.52], $p$-value = $3.00 \times 10^{-9}$; two-sided Fisher's exact test). In the Australian validation cohort[31], the majority (81/85, 95.3%) of *mtrA* LOF strains had concurrent *mtrCDE* overexpression mutations, so it was not possible to test for these associations (Supplementary Table 11). In such genetic backgrounds where overexpression mutations mask the effect of *mtrA* LOF, *mtrC* LOF is the preferred method of efflux pump downregulation: the majority of *mtrC* LOF mutations in both the global dataset (174/180, 96.7%) and the Australian cohort (33/35, 94.3%) occurred in backgrounds with known *mtr* overexpression mutations (Supplementary Tables 12–13). Phylogenetic analysis showed that the

distribution of *mtrA* LOF differed from that of *mtrC* LOF with fewer introductions but more sustained transmission and that the two mutations were largely non-overlapping (Supplementary Fig. 3); only four strains had both *mtrA* and *mtrC* LOF mutations. Our results indicate that multiple adaptive paths for MtrCDE efflux pump downregulation exist depending on genetic interactions with other concurrent mutations in the *mtrCDE* regulon.

**FarAB efflux pump LOF is associated with cervical infection.** The associations we observed in the *mtrCDE* regulon raised the question of the mechanism by which the cervical environment could select for pump downregulation. Recent work on *Pseudomonas* suggested one possible model: overexpression of homologous *P. aeruginosa* efflux pumps belonging to the same resistance/nodulation/cell division (RND) proton/substrate antiporter family as MtrCDE results in a fitness cost due to increased cytoplasmic acidification[34]. This fitness cost was only observed in anaerobic conditions, where aerobic respiration cannot be used to dissipate excess protons efficiently[34]. Analogous conditions in the female urogenital tract, potentially augmented by environmental acidity, could create a similar selective pressure during human infection that leads to pump downregulation or loss.

This model predicts that adaptation to these conditions would similarly result in the downregulation of FarAB, the other proton-substrate antiporter efflux pump in *N. gonorrhoeae*. FarAB is a member of the major facilitator superfamily (MFS) of efflux pumps and effluxes long-chain fatty acids[35,36]. In our global

dataset, 332/4838 (6.86%) of isolates were predicted to have *farA* LOF mutations, of which the majority (316/332; 95.2%) were due to indels in a homopolymeric stretch of eight 'T' nucleotides near the 5' end of the gene (Supplementary Fig. 2C). *farA* LOF mutations were associated with patient sexual behavior (*p*-value = 5.06 × 10⁻¹⁰; two-sided Fisher's exact test) and site of infection (*p*-value = 1.78 × 10⁻¹²; two-sided Fisher's exact test) and over-represented in cervical isolates: 33 out of 129 (25.6%) cervical isolates contained a *farA* LOF mutation compared to 117 out of 2246 urethral isolates (5.21%; OR = 6.25, 95% CI [3.90–9.83], *p*-value = 3.24 × 10⁻¹³; two-sided Fisher's exact test), 3 out of 106 pharyngeal isolates (2.83%; OR = 11.70, 95% CI [3.50–61.61], *p*-value = 3.80 × 10⁻⁷; two-sided Fisher's exact test), and 12 out of 246 rectal isolates (4.88%; OR = 6.66, 95% CI [3.19-14.80], *p*-value = 1.57 × 10⁻⁸; two-sided Fisher's exact test) (Fig. 3, Supplementary Table 14 and 15). *farA* LOF mutations were prevalent also in our Australian validation dataset[31] (225/2180; 10.32%) and again associated with sexual behavior (*p*-value < 2.20 × 10⁻¹⁶; two-sided Fisher's exact test) and site of infection (*p*-value < 2.20 × 10⁻¹⁶; two-sided Fisher's exact test) (Supplementary Fig. 5, Supplementary Tables 16 and 17). The phylogenetic distribution of *farA* LOF indicated sustained transmission (Supplementary Fig. 3) and overlapped with that of *mtrA* LOF mutations (48.9% of isolates with *mtrA* LOF mutations also had *farA* LOF mutations), potentially indicating additive contributions to cervical adaptation. Furthermore, MtrR activates *farAB* expression by repressing the *farR* repressor[37]. This cross-talk between the two efflux pump operons indicates that in *mtrCDE* overexpression strains where MtrR activity is impaired, the effect of *farA* LOF—like *mtrA* LOF—may be masked. We did not observe frequent LOF mutations in the sodium gradient-dependent MATE family efflux pump NorM[38] or in the ATP-dependent ABC family pump MacAB[39] (Supplementary Table 3). The prevalence and cervical enrichment of *farA* LOF mutations and the relative rarity of LOF mutations in other non-proton motive force-driven pumps suggest that cytoplasmic acidification may be a mechanism by which the female urogenital tract selects for efflux pump loss.

**Meningococcal *mtrC* LOF is driven by urogenital adaptation.** *N. meningitidis*, a species closely related to *N. gonorrhoeae*, colonizes the oropharyngeal tract and can cause invasive disease, including meningitis and septicemia[40]. We characterized *mtrC* diversity in a collection of 14,798 *N. meningitidis* genomes, reasoning that the cervical environmental pressures that select for efflux pump LOF in the gonococcus will be rarely encountered by the meningococcus. In agreement with this, the 'GC' hexarepeat associated with most gonococcal *mtrC* LOF mutations was less conserved in *N. meningitidis*; only 9684/14798 (65.4%) isolates contained an intact hexarepeat compared to 4644/4847 (95.8%) of *N. gonorrhoeae* isolates (*p*-value < 2.2 × 10⁻¹⁶; two-sided Fisher's exact test). In this same collection, we observed *mtrC* LOF due to deletions in the hexarepeat region in only 82 meningococcal isolates (0.55%), with a similar frequency of 25/4059 (0.62%) in a curated dataset comprising all invasive meningococcal disease isolates collected in the UK from 2009 to 2013[41]. The observed interruption of 'GC' dinucleotide repeats, predicted to result in a lower mutation rate[42], and the relative rarity of *mtrC* LOF mutations suggests that efflux pump loss is not generally adaptive in *N. meningitidis*. However, a urogenitally-adapted meningococcal lineage has recently emerged in the US associated with outbreaks of non-gonococcal urethritis in heterosexual patients[43–45]. In isolates from this lineage, the prevalence of *mtrC* LOF mutations was 18/207 (8.70%), substantially higher than typical *N. meningitidis* and comparable to the prevalence of

gonococcal *mtrC* LOF mutations in MSW in our global dataset (4.47%). We compared the frequency of *mtrC* LOF mutations in the urogenital lineage to geographically and genetically matched isolates (i.e., all publicly available *n* = 456 PubMLST ST-11 North American isolates) and observed a significant difference in prevalence (18 out of 207 or 8.70% versus 2 out of 249 or 0.80%; OR = 11.71, 95% CI [2.75–105.37], *p*-value = 3.31 × 10⁻⁵; two-sided Fisher's exact test). Most *mtrC* LOF mutations occurred due to the same hexarepeat two base pair deletion that we previously observed for *N. gonorrhoeae*, and in line with this, *mtrC* LOF in this urogenital lineage arose multiple times independently similarly to gonococcal *mtrC* LOF mutations (Fig. 4, Supplementary Fig. 3). *farA* LOF mutations were not observed in this meningococcal lineage. We conclude that MtrCDE efflux pump LOF is rare in typical meningococcal strains that inhabit the oropharynx but elevated in frequency in a unique urogenitally-adapted lineage circulating in heterosexuals, indicative of potential ongoing adaptation to the cervical niche. Our results suggest that efflux pump loss is broadly adaptive for cervical colonization across pathogenic *Neisseria*.

**Discussion**

In an era in which widespread antimicrobial pressure has led to the emergence of extensively drug-resistant *N. gonorrhoeae*[46], isolates that appear to have reverted to susceptibility still arise[9,10], demonstrating that antibiotic and host environmental pressures interact to shape the evolution of *N. gonorrhoeae*. Here, we showed that frameshift-mediated truncations in the *mtrC* component of the MtrCDE efflux pump are the primary mechanism for epistatic increases in antibiotic susceptibility across a global collection of clinical gonococcal isolates, as suggested by prior work[20,22]. *mtrC* LOF mutations are enriched in cervical isolates and a frameshifted form of the pump activator MtrA exhibits similar trends, supporting a model in which reduced or eliminated *mtrCDE* efflux pump expression contributes to adaptation to the female genital tract. We hypothesized that the mechanism by which this occurs is through increased cytoplasmic acidification in anaerobic conditions[34] and demonstrated that LOF mutations in *farA*, encoding a subunit of the other proton motive force-driven pump FarAB, were likewise enriched in cervical isolates. The LOF mutations we observed in *mtrC* and *farA* primarily occurred in short homopolymeric sequences (though with low numbers of repeated units) and thus may occur at higher rates than insertions or deletions in non-repetitive regions or nonsense mutations, similar to other resistance suppressor mutations[47], though this will need to be confirmed in future experiments. In total, 42.6% of cervical isolates in the global dataset and 32.6% in the validation dataset contained a LOF mutation in either *mtrC, farA,* or *mtrA*, indicating that efflux pump downregulation via multiple genetic mechanisms is prevalent in cervical infection. These results complement prior studies suggesting that *mtrR* LOF resulting in increased resistance to fecal lipids plays a critical role in gonococcal adaptation to the rectal environment[48,49] and taken together suggest a model in which the fitness benefit of efflux pump expression is highly context dependent.

Other selective forces could also have contributed to the observed enrichment of LOF mutations in cervical isolates. For instance, iron levels modulate *mtrCDE* expression through Fur (the ferric uptake regulator) and MpeR[50]. Iron limitation results in increased expression of *mtrCDE*, and conversely, iron enrichment result in decreased expression, suggesting a fitness cost for *mtrCDE* expression during high iron conditions. Variation in environmental iron levels, such as in the menstrual cycle, may provide another selective pressure for LOF mutations to arise

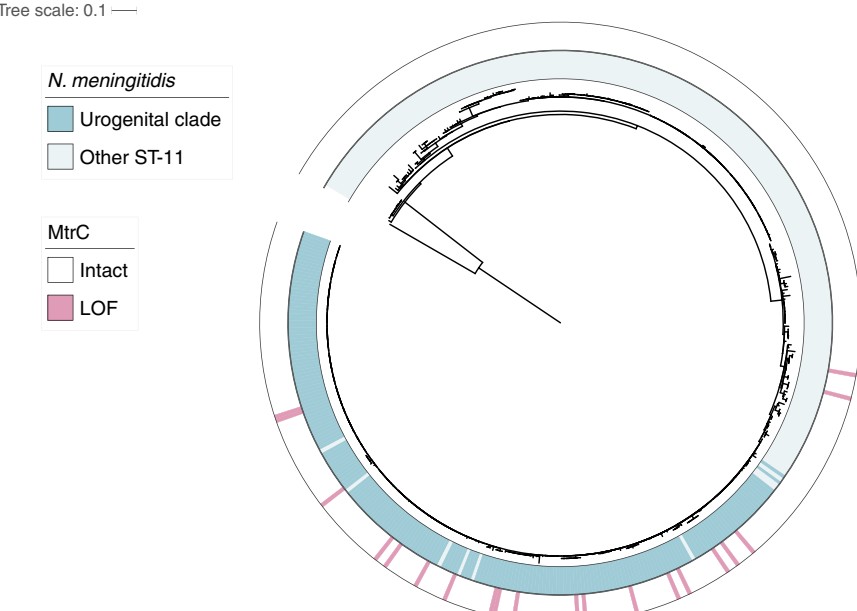

Tree scale: 0.1 ─

**Fig. 4 *mtrC* LOF mutations are enriched in a lineage of ST-11 urogenitally-adapted *N. meningitidis*.** A core-genome maximum likelihood phylogeny based on 25045 SNPs was estimated of all North American ST-11 *N. meningitidis* strains from PubMLST (*n* = 456; accessed 2019–09–03) rooted with meningococcal reference genome MC58. Membership in the ST-11 urogenital clade (dark blue) is defined as in Retchless et al.[44]. Genomes with *mtrC* LOF mutations are indicated in pink. Branch length represents substitutions per site.

particularly when MtrR function is impaired through active site or promoter mutations. Differing rates of antibiotic use for gonorrhea in men and women due to increased asymptomatic infection in women might also select for *mtrC* LOF mutations, but this would not explain the associations we observed for the non-antibiotic substrate efflux pump *farAB* or the increased frequency of *mtrC* LOF mutations in urogenitally-adapted meningococci. RNA sequencing from men and women infected with gonorrhea demonstrated a 4-fold lower expression of *mtrCDE* in women, re-affirming the idea that efflux pump expression in the female genital tract incurs a fitness cost[51].

Despite significant associations, only a proportion of cervical isolates exhibited these LOF genotypes, suggesting variation in cervix-associated pressures or indicating that cervical culture specimens were obtained before niche pressures could select for pump downregulation. This variation could also lead to mixed populations of efflux pump wild-type and LOF strains; however, because only one clonal isolate per site per patient is typically sequenced in clinical surveillance studies, we would be unable to detect this intra-host patient diversity. Targeted amplicon sequencing of LOF loci directly from patients in future studies would help to assess whether this intra-host diversity plays a role in infection and transmission. In particular, this intra-host pathogen diversity could facilitate transmission from the female genital tract to other sites of infection, where efflux pump activity incurs less of a fitness cost. In those new sites, isolates with wild-type efflux pump loci in the mixed population could selectively expand relative to LOF efflux pump strains and also serve as possible recombination donors of wild-type alleles. This standing genetic variation would, therefore, facilitate gonococcal adaptation across different mucosal niches. Additionally, while the cervix is the primary site of infection and source for culture in women, the selective pressures at play may include other sites more broadly in the female genital tract and may be influenced by the presence of other microbial species both pathogenic and commensal.

Our model extended to the other pathogenic *Neisseria* species, *N. meningitidis*, in that a urogenital clade transmitting in

primarily heterosexual populations appeared to be undergoing further urogenital adaptation via the same *mtrC* frameshift mutation that was most commonly observed for *N. gonorrhoeae*. In the absence of data on cases of cervicitis, we hypothesized that for this meningococcal lineage, efflux pump LOF emerged in the female urogenital tract and was transmitted to heterosexual men resulting in the enrichment we observed. Efflux pumps are common across Gram-negative bacteria[52], and their loss may be a general adaptive strategy for species that face similar pressures as *N. gonorrhoeae* and urogenitally-adapted *N. meningitidis*. In support of this, clinical isolates of *Pseudomonas aeruginosa* with truncations in genes homologous to *mtrC*[53,54] and exhibiting antibiotic hypersensitivity have been obtained from cystic fibrosis patients, in whom the thick mucus in airway environments can likewise exhibit increased acidity and decreased oxygen availability[55,56].

Our results also suggest potential therapeutic avenues for addressing the emergence of multidrug-resistant gonococcal strains. Selective knockdown of MtrCDE homologs in other bacteria via antisense RNA[57] and bacteriophages[58] has successfully re-sensitized resistant strains and enhanced antibiotic efficacy, and ectopic expression in *N. gonorrhoeae* of the *mtrR* repressor in a cephalosporin-resistant strain enhances gonococcal killing by β-lactam antibiotics in the mouse model[59]. Our population-wide estimated effect sizes for *mtrC* LOF mutations provide a prediction for the re-sensitization effect of MtrCDE knockdown across multiple genetic backgrounds and suggest particularly strong effects for the macrolide azithromycin (Supplementary Fig. 4). Because the correlation between MIC differences and clinical efficacy is still not well understood[60,61], follow up studies to assess treatment efficacy differences in patients with and without *mtrC* LOF strains can help to quantify the expected effect of MtrCDE knockdown in the clinical context.

In summary, by analysis of population genomics and patient clinical data, we have shown that pathogenic *Neisseria* can use multiple avenues of efflux pump perturbation as an adaptive strategy to respond to host environmental pressures and illustrate

how these host pressures may result in increased antibiotic susceptibility in *N. gonorrhoeae*.

## Methods

**Genomics pipeline.** Reads for isolates with either associated azithromycin, ciprofloxacin, or ceftriaxone MIC metadata were downloaded from datasets listed in Supplementary Table 1. Reads were inspected using FastQC version 0.11.7 (https://www.bioinformatics.babraham.ac.uk/projects/fastqc/) and removed if GC content diverged notably from expected values (~52–54%) or if base quality was systematically poor. We mapped read data to the NCCP11945 reference genome (RefSeq accession: NC_011035.1) using BWA-MEM (version 0.7.17-r1188)[62,63] and deduplicated reads using Picard (version 2.8.0) (https://github.com/broadinstitute/picard). BamQC in Qualimap (version 2.2.1)[64] was run to identify samples with less than 70% of reads aligned or samples with less than 40X coverage, which were discarded. We used Pilon (version 1.16)[65] to call variants with mindepth set to 10 and minmq set to 20 and generated pseudogenomes from Pilon VCFs by including all PASS sites and alternate alleles with AF > 0.9; all other sites were assigned as 'N'. Samples with greater than 15% of sites across the genome missing were also excluded. We created de novo assemblies using SPAdes (version 3.12.0 run using 8 threads, paired end reads where available, and the --careful flag set)[66] and quality filtered contigs to ensure coverage greater than 10X, length greater than 500 base pairs, and total genome size approximately equal to the FA1090 genome size (2.0–2.3 Mbp). We annotated assemblies with Prokka (version 1.13)[67], and clustered core genes using Roary (version 3.12)[68] (flags -z -e -n -v -s -i 92) and core intergenic regions using piggy (version 1.2)[69]. A recombination-corrected phylogeny of all isolates was constructed by running Gubbins (version 2.3.4) on the aligned pseudogenomes and visualized in iTOL (version 4.4.2)[70–72]. All isolates with associated metadata and accession numbers are listed in Supplementary Data 3 and 4.

**Resistance allele calling.** Known resistance determinants in single-copy genes were called by identifying expected SNPs in the pseudogenomes. For categorizing mosaic alleles of *mtr*, we ran BLASTn (version 2.6.0)[73] on the de novo assemblies using a query sequence from FA1090 (Genbank accession: NC_002946.2) comprising the *mtr* intergenic promoter region and *mtrCDE*. BLAST results were aligned using MAFFT (version 7.450)[74] and clustered into distinct allelic families using FastBAPS (version 1.0.0)[75]. We confirmed that horizontally-transferred *mtr* alleles associated with resistance from prior studies[5] corresponded to distinct clusters in FastBAPS. A similar approach was used to cluster *penA* alleles after running BLASTn with a *penA* reference sequence from FA1090. Variant calling in the multi-copy 23S rRNA locus was done by mapping to a modified NCCP11945 reference genome containing only one copy of the 23S rRNA and analyzing variant allele frequencies[76]. We identified truncated MtrR proteins using Prokka annotations, and mutations in the *mtr* promoter region associated with upregulation of *mtrCDE* (A deletion and TT insertion in inverted repeat, *mtr* 120) using an alignment of the *mtr* promoter from piggy output.

**Phenotype processing and linear models.** We doubled GISP azithromycin MICs before 2005 to account for the GISP MIC protocol testing change[77]. Samples with binary resistance phenotypes (i.e., "SUS" and "RES") were discarded. For samples with MICs listed as above or below a threshold (indicated by greater than or less than symbols), the MIC was set to equal the provided threshold. MICs were $\log_2$-transformed for use as continuous outcome variables in linear modeling and GWAS. We modeled transformed MICs using a panel of known resistance markers[11,12] and included the recently characterized mosaic *mtrCDE* alleles[5] and *rplD* G70D substitution[78] conferring azithromycin resistance, as well as isolate country of origin. Formulas called by the lm function in R (version 3.5.1) for each drug were (with codon or nucleotide site indicated after each gene or rRNA, respectively):

Azithromycin: Log_AZI ~ Country + MtrR 39 + MtrR 45 + MtrR LOF + *mtrR* promoter + *mtrCDE* BAPS + RplD G70D + 23S rRNA 2059 + 23S rRNA 2611.

Ceftriaxone: Log_CRO ~ Country + MtrR 39 + MtrR 45 + MtrR LOF + *mtrR* promoter + *penA* BAPS + PonA 421 + PenA 501 + PenA 542 + PenA 551 + PorB 120 + PorB 121.

Ciprofloxacin: Log_CIP ~ Country + MtrR 39 + MtrR 45 + MtrR LOF + *mtrR* promoter + GyrA 91 + GyrA 95 + ParC 86 + ParC 87 + ParC 91 + PorB 120 + PorB 121.

To visualize the continuous MICs using thresholds as on Fig. 1, we binarized MICs using the CLSI resistance breakpoint for ciprofloxacin, the CLSI non-susceptibility breakpoint for azithromycin, and the CDC GISP surveillance breakpoint for ceftriaxone.

**GWAS and unitig annotation.** We used a regression-based GWAS approach to identify novel susceptibility mutations. In particular, we employed a linear mixed model with a random effect to control for the confounding influence of population structure and a fixed effect to control for isolate country of origin. Though the outcome variable ($\log_2$-transformed MICs) is the same, in contrast to the linear modeling approach described above, which models the linear, additive effect of multiple, known resistance mutations, regression in a GWAS is usually run

independently and univariately on each variant for all identified variants in the genome, providing a systematic way to identify novel contributors to the outcome variable. Linear mixed model GWAS was run using Pyseer (version 1.2.0 with default allele frequency filters) on the 480,902 unitigs generated from GATB (version 1.3.0); the recombination-corrected phylogeny from Gubbins was used to parameterize the Pyseer population structure random effects term and isolate country of origin was included as a fixed effect covariate. To create the Manhattan plot, we mapped all unitigs from the GWAS using BWA-MEM (modified parameters: -B 2 and -O 3) to the pan-susceptible WHO F strain reference genome (Genbank accession: GCA_900087635.2) edited to contain only one locus of the 23S rRNA. Significant unitigs were annotated using Pyseer's annotation pipeline. Unitigs mapping to multiple sites in the genome and in or near the highly variable *pilE* (encoding pilus subunit) or *piiC* (encoding opacity protein family) genes were excluded, as were unitigs less than twenty base pairs in length. Due to redundancy and linkage, variants will be spanned by multiple overlapping unitigs with similar frequencies and *p*-values. For ease of interpretation, we grouped unitigs within 50 base pairs of each other and represented each cluster by the most significant unitig. Unitigs with allele frequency greater than 50% were also excluded as they represented the majority allele. Unitig clusters were then annotated by gene or adjacent genes for unitigs mapping to intergenic regions and further analyzed for predicted functional effect relative to the WHO F reference genome in Geneious Prime (version 2019.2.1, https://www.geneious.com).

**Identifying LOF and upregulation alleles.** To identify predicted LOF alleles of efflux pump proteins, we ran BLASTn on the de novo assemblies using a query sequence from FA1090 (reference genome FA19 was used for *mtrA*). Sequences that were full-length or approximately full-length ($+/-5$ nucleotides) beginning with expected start codons were translated using Python (version 3.6.5) and Biopython (version 1.69)[79]. Peptides shorter than 90% of the expected full-length size of the protein were further analyzed using Geneious Prime (version 2019.2.1, https://www.geneious.com) to identify the nucleotide mutations resulting in predicted LOF by alignment of the nucleotide sequences. We called *mtrCDE* overexpression status by identifying the presence of any of the known *mtrR* promoter mutations, MtrR coding sequence mutations, and mosaic *mtrCDE* alleles.

**Experimental validation.** *N. gonorrhoeae* culture was conducted on GCB agar (Difco) plates supplemented with 1% Kellogg's supplements[80] at 37 °C in a 5% $CO_2$ atmosphere. Antimicrobial susceptibility testing was conducted on GCB agar supplemented with 1% IsoVitaleX (Becton Dickinson) using Etests (bioMérieux) at 37 °C in a 5% $CO_2$ atmosphere. We selected a clinical isolate (NY0195[81]) from the multidrug-resistant lineage ST-1901[23] that contained an *mtrC* LOF mutation mediated by a two base pair hexarepeat deletion and confirmed via Etests that its MIC matched, within one dilution, its reported MIC. Isolate NY0195 contained mosaic *penA* allele XXXIV conferring cephalosporin reduced susceptibility and the *gyrA* S91F substitution conferring ciprofloxacin resistance[9]. We complemented the *mtrC* LOF mutation in this strain by transforming it via electroporation[80] with a 2 kb PCR product containing a *Neisserial* DNA uptake sequence and an in-frame *mtrC* allele, obtained by colony PCR from a neighboring isolate (GCGS0759). After obtaining transformants by selecting on an azithromycin 0.05 µg/mL GCB plate supplemented with Kellogg's supplement, we confirmed successful transformation by Sanger sequencing of the *mtrC* gene. No spontaneous mutants on azithromycin 0.05 µg/mL plates were observed after conducting control transformations in the absence of GCGS0759 *mtrC* PCR product. We conducted antimicrobial susceptibility testing in triplicate using Etests, assessing statistical significance between parental and transformant MICs by a two-sample *t*-test.

**Metadata analysis.** Patient metadata were collected from the following publications from Supplementary Table 1 that had information on site of infection: Demczuk et al.[82], Demczuk et al.[83], Ezewudo et al.[84], Grad et al.[85], Grad et al.[9], Kwong et al.[86], Lee et al.[15], and Mortimer et al.[81]. Sites of infection were standardized across datasets using a common ontology (i.e., specified as urethra, rectum, pharynx, cervix, or other). Two-sided Fisher's exact test in R (version 3.5.1) was used to infer whether there was nonrandom association between *mtrC* LOF presence and either anatomical site of infection or sexual behavior. For sexual behavior analysis, isolates cultured from multiple sites on the same patient were counted as only one data point.

**Meningococcal *mtrC* analysis.** *mtrC* alleles from *N. meningitidis* assembled genomes were downloaded from PubMLST ($n = 14798$; accessed 2019–09–03) by setting (species = "Neisseria meningitidis"), filtering by (Sequence bin size >= 2 Mbp), and exporting sequences for Locus "NEIS1634"[87]. *mtrC* LOF alleles were identified as described above. We generated a core-genome maximum likelihood phylogeny of all North American ST-11 *N. meningitidis* strains from PubMLST ($n = 456$; accessed 2019–09–03) rooted with meningococcal reference genome MC58 (Genbank accession: AE002098.2) using Roary (version 3.12) (flags -z -e -n -v -s -i 92) and annotated it using metadata from Retchless et al.[44] (see Supplementary Data 5 for PubMLST IDs). Overrepresentation of *mtrC* LOF alleles in the US urogenital lineage compared to selected control datasets was assessed using two-sided Fisher's exact test in R (version 3.5.1).

**Reporting summary**. Further information on research design is available in the Nature Research Reporting Summary linked to this article.

## Data availability

In Supplementary Data 3–4, we have included accession numbers (via publicly hosted database NCBI SRA) for accessing all raw sequence data used for *N. gonorrhoeae* analyses. Intermediate outputs from the genomics pipeline (e.g., de novo assemblies) may also be available from the authors upon request. In Supplementary Data 5, we have included accession numbers (via publicly hosted database PubMLST: https://pubmlst.org/neisseria/) for accessing all sequence data used for *N. meningitidis* analyses. Source data underlying all figures are available in Supplementary Data 1–2 or at https://github.com/gradlab/mtrC-GWAS.

## Code availability

Code to reproduce the analyses and figures is available at https://github.com/gradlab/mtrC-GWAS or from the authors upon request.

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

## Acknowledgements

This work was supported by the NIH/NIAID grant 1R01AI132606-01 and the Smith Family Foundation. T.D.M. is additionally supported by the NIH/NIAID F32AI145157, and K.C.M. is additionally supported by the NSF GRFP. D.H.F.R. was supported by award Number T32GM007753 from the National Institute of General Medical Sciences. The content is solely the responsibility of the authors and does not necessarily represent the official views of the National Institute of General Medical Sciences or the National Institutes of Health. D.A.W. is supported by an Early Career Fellowship from the National Health and Medical Research Council of Australia (GNT1123854). Portions of this research were conducted on the O2 high-performance computing cluster, supported by the Research Computing Group at Harvard Medical School. This publication made use of the Meningitis Research Foundation Meningococcus Genome Library (http://www.meningitis.org/research/genome) developed by Public Health England, the Wellcome Trust Sanger Institute, and the University of Oxford as a collaboration and funded by the Meningitis Research Foundation. The authors additionally thank Crista Wadsworth, Samantha Palace, and other members of the Grad Lab for helpful comments during development of the project.

## Author contributions

K.C.M., T.D.M., A.L.H., N.E.W., and L.S.B. performed and interpreted genomic analyses. K.C.M., D.H.F.R., and Y.W. performed experimental analyses. D.G. and M.U. provided data and conducted genomic analyses on historical isolates. G.T. and D.A.W. provided data and interpreted results for the validation dataset. S.R.H., M.U., D.A.W., and Y.H.G. supervised the project. K.C.M., T.D.M., and Y.H.G. wrote the paper with contributions from all authors.

## Competing interests

The authors declare no competing interests.
