## [Peer Review File · Nature Communications]

REVIEWER COMMENTS

Reviewer #1 (Remarks to the Author):

Ma et al January 2020

This is a very interesting manuscript that analyzes the genomic sequences of many *Neisseria gonorrhoeae* isolates from different studies to show that a high proportion of cervical isolates have mutations that disrupt the two main efflux pump systems, but that the proportion of isolates from other sites have lower proportions of efflux-deficient isolates. While I do not have the expertise to evaluate their methods or statistics; if these are appropriate, this is a novel observation. The text and most of the data are clearly presented. The discussion is short and doesn't provide sufficient speculation on why there might be a higher proportion of efflux mutants in cervical isolates. I would prefer that the authors provide a bit more speculation about selective forces.

Comments:

1. While the authors have speculated that a slipped-strand mispairing mechanism is behind the loss of two nucleotides, the rate of slippage of this hexanucleotide sequence would be expected to be very low under normal conditions. Have the authors examined other potential phase variable genes in these genomes to test whether the rates of slippage or other mechanism is increased during cervical colonization? Regardless of these analyses, it would be helpful if other environmental selective forces, beside the proposed cytoplasmic acidification shown in *Pseudomonas*, were explored.
2. The hexanucleotide site that changes could be a recognition site for a restriction enzyme (e.g., BssHI) or more importantly a modification methylase. Why such an activity has not to my knowledge been described in the *Neisseria*, it would be good to analyze the genomic sequences for genes that could have such an activity. While it is not possible to always predict the DNA sequence recognized by these enzymes, if such an activity is involved, certain related genes would correlate with the deletion.
3. It is not clear from the text whether any of the efflux deficient isolates caused a treatment failure.
4. Clinical laboratories only save one isolate from a patient and it would be that isolate that was sequenced. There is a possibility the authors might want to consider that these cervical isolates were co-colonized with an efflux competent clone.
5. It was not obvious to me whether Mtr and FarR mutations were found in the same isolates or not. A comment about this and how that relates to phenotype would be useful.
6. Past literature has speculated that efflux might be most important in the rectum due to fatty acids and other toxic lipids. Perhaps this should be discussed.
7. Supplementary Figure 1, Supplementary Figure 2 legends need more explanations of the figures.

Reviewer #2 (Remarks to the Author):

The article by Ma et al makes an important, novel contribution to knowledge regarding antimicrobial resistance mechanisms in *Neisseria gonorrhoeae* (Ng). The main analysis method was bioinformatics –using a global dataset comprising 4882 genomes to perform genome wide association to identify loss-of-function (LOF) mutations in the MtrCDE efflux pump – i.e. to search for mutations that would cause decreased efflux. The research question in this research is very important - why are some clinical isolates of Ng more susceptible to antibiotics? Quantitative predictions of MICs were based on a well-established method using multiple regression on a panel of resistance markers. The authors identified a loss of function (LOF) mutation in the MtrCDE efflux pump (in *mtrC*) associated with increased susceptibility and the same variant was associated with increased susceptibility to ceftriaxone, ciprofloxacin and other antibiotics. The authors also noted additional LOF mutations associated with increased susceptibility. They also show that the acquisition of antimicrobial susceptibility based on the mutation in *mtrC* was “relatively common” and globally prevalent. Another surprising finding was that the *mtrC* LOF was more prevalent in isolates from men who have sex with women as compared to men who have sex with men, and that this mutation was also more prevalent in cervical isolates. This later finding was validated using a different data set. The authors conclude that environmental pressures in the female reproductive tract may contribute to such a selection. The

authors also determined the same associations for isolates with mtrA LOF mutations. The authors next hypothesized that environmental acidity could create elective pressure leading to loss of function of the efflux pump. Further analysis with FarAB indicated that mutations in farA caused similar associations – i.e. with patient sex of sex partner and cervical isolates. Finally, the authors reported that mtrC LOF mutations were enriched in a heterosexual urethritis-associated lineage of *N. meningitidis*.

Minor comments:

1. For some data, accession numbers of the data are not provided (e.g.
 - a. *N. meningitidis* ST-11, North America)
2. Perhaps the term “unitig” could be better described.
3. Parameters for assembly generation (line421) could be indicated
4. Line 510 – “supplementary tables” used should be clarified.
5. Can the authors explain why farA and mtrA LOFs tended to coincide rather than farA and mtrC LOFs (Supplementary Figure 3).
6. It is difficult to read the parts of Figure 2. Would the authors consider adding the regression coefficients for each antibiotic to data in supplementary Table 5? This would be valuable information for investigators who might refer to this research.
7. Line 486 – What is Geneious Prime (software available?)
8. A discussion of the advantages of GWAS over the multiple regression model might be informative for the uninitiated (i.e. perhaps lines 128-131 could be expanded). Perhaps the authors could address machine learning vs GWAS as an analytical tool?
9. The biological relevance of these mutations could be discussed – the numbers of genomes with these mutations was (relatively) small. Would the authors care to comment on whether such mutations make a difference in how women respond to antibiotics?

Reviewer comments

Reviewer #1 (Remarks to the Author):

Ma et al January 2020

This is a very interesting manuscript that analyzes the genomic sequences a many *Neisseria gonorrhoeae* isolates from different studies to show that a high proportion of cervical isolates have mutations that disrupt the two main efflux pump systems, but that the proportion of isolates from others sites have lower proportions of efflux-deficient isolates. While I do not have the expertise to evaluate their methods or statistics; if these are appropriate, this is a novel observation. The text and most of the data are clearly presented. The discussion is short and doesn't provide sufficient speculation on why there might be a higher proportion of efflux mutants in cervical isolates. I would prefer that the authors provide a bit more speculation about selective forces.

We thank the reviewer for their helpful comments and summary of our manuscript. We agree that additional description of other selective forces that could be responsible for the observed efflux pump LOF mutations would be helpful. We have added the following points to the discussion outlining other possible models: **Lines 392-406** "Other selective forces could also have contributed to the observed enrichment of LOF mutations in cervical isolates. For instance, iron levels modulate *mtrCDE* expression through Fur (the ferric uptake regulator) and MpeR⁵⁷. Iron limitation results in increased expression of *mtrCDE*, and iron enrichment result in decreased expression, suggesting a fitness cost for *mtrCDE* expression during high iron conditions. Variation in environmental iron levels, such as in the menstrual cycle, may provide another selective pressure for LOF mutations, particularly when MtrR function is impaired through active site or promoter mutations. Differing rates of antibiotic use for gonorrhea in men and women due to increased asymptomatic infection in women might also select for *mtrC* LOF mutations, but this would not explain the associations we observed for the non-antibiotic substrate efflux pump *farAB* or the increased frequency of *mtrC* LOF mutations in urogenitally-adapted meningococci. RNA sequencing from men and women infected with gonorrhea demonstrated a 4-fold lower expression of *mtrCDE* in women, re-affirming the idea that efflux pump expression in the female genital tract incurs a fitness cost⁵⁸."

Comments:

1. While the authors have speculated that a slipped-strand mispairing mechanism is behind the loss of two nucleotides, the rate of slippage of this hexanucleotide sequence would be expected to be very low under normal conditions. Have the authors examined other potential phase variable genes in these genomes to test whether the rates of slippage or other mechanism is increased during cervical colonization? Regardless of these analyses, it would be helpful if other environmental selective forces, beside the proposed cytoplasmic acidification shown in *Pseudomonas*, were explored.

We agree that the rate of slippage is expected to be lower than canonical phase variable genes due to the number of repeats, but we think the rate could potentially be higher than the mutation rate for a SNP or insertion/deletion in a non-repetitive region. We have re-worded this section accordingly: **Lines 171-172** "mtrC LOF mutations have emerged numerous times throughout the phylogeny (Supplementary Figure 3), indicative of possible repeated losses of a dinucleotide in the hexarepeat region due to DNA polymerase slippage, which may occur at a higher rate than single nucleotide nonsense mutations."

To see if overall slippage rate was elevated in the cervical environment, we looked at the phase variation rate in other genes with GC dinucleotide hexarepeats. We found two additional genes, *pdxA* (predicted to encode a 4-hydroxythreonin-4-phosphate dehydrogenase) and *lpxC* (predicted to

encode a UDP-3-O-acyl-N-acetylglucosamine deacetylase), that also contain 6 GC repeats. However, frameshift mutations were not present in either gene in our dataset across all sites of infection. A previous TnSeq study (Remmele et al., 2014) classified *lpxC* as an essential gene, which would contribute to the absence of observed frameshifts. The essentiality of *pdxA* is unclear. Thus, we were unable to show an elevated rate of slippage during cervical colonization.

2. The hexanucleotide site that changes could be a recognition site for a restriction enzyme (e.g., BssHI) or more importantly a modification methylase. Why such an activity has not to my knowledge been described in the Neisseria, it would be good to analyze the genomic sequences for genes that could have such an activity. While it is not possible to always predict the DNA sequence recognized by these enzymes, if such an activity is involved, certain related genes would correlate with the deletion.

This is a good suggestion and timely as just recently, the methylation targets of the gonococcal methylases have been identified using SMRT sequencing (Sanchez-Buso et al., 2019). We checked the recognition sites identified in that publication (in Table 1: <https://www.nature.com/articles/s41598-019-51102-2/tables/1>) but none of them corresponded to the 'GC' dinucleotide repeat region or its direct neighboring sequences. We have mentioned these results in the text: **Lines 138-140** "We also checked whether the two base pair deletion would affect recognition by any of the gonococcal methylases³², but no methylase target motif sites mapped to the hexarepeat or its direct surrounding sequences."

3. It is not clear from the text whether any of the efflux deficient isolates caused a treatment failure.

While we did not have access to patient treatment failure data, we believe that at least for the *mtrC* LOF strains exhibiting increased antibiotic susceptibility, antibiotic treatment would be increased in efficacy. We have expanded upon this point in the discussion, particularly in the context of therapeutic options to knockout *mtrCDE*: **Lines 440-446** "Our population-wide estimated effect sizes for *mtrC* LOF mutations provide a prediction for the re-sensitization effect of MtrCDE knockdown across multiple genetic backgrounds and suggest particularly strong effects for the macrolide azithromycin (Supplementary Figure 4). Because the correlation between MIC differences and clinical efficacy is still not well understood^{72,73}, follow up studies to assess treatment efficacy differences in patients with and without *mtrC* LOF strains can help to quantify the expected effect of MtrCDE knockdown in the clinical context."

4. Clinical laboratories only save one isolate from a patient and it would be that isolate that was sequenced. There is a possibility the authors might want to consider that these cervical isolates were co-colonized with a efflux competent clone.

We think this intra-patient diversity could indeed be interesting to explore and have added this as a limitation to our study in the discussion: **Lines 408-415** "Despite significant associations, only a proportion of cervical isolates exhibited these LOF genotypes, suggesting variation in cervix-associated pressures or indicating that cervical culture specimens were obtained before niche pressures could select for pump downregulation. This variation could also lead to mixed populations of efflux pump WT and LOF strains; however, because only one clonal isolate per site per patient is typically sequenced in clinical surveillance studies, we would be unable to detect this intra-host patient diversity. Targeted amplicon sequencing of LOF loci directly from patients in future studies would help to assess whether this intra-host diversity plays a role in infection and transmission."

5. It was not obvious to me whether Mtr and FarR mutations were found in the same isolates or not. A comment about this and how that relates to phenotype would be useful.

We describe the overlap between *mtrA* and *farA* LOF mutations in greater detail: **Lines 311-317** “The phylogenetic distribution of *farA* LOF indicated sustained transmission (Supplementary Figure 3) and overlapped with that of *mtrA* LOF mutations, potentially indicating additive contributions to cervical adaptation. Furthermore, MtrR activates *farAB* expression by repressing the *farR* repressor⁴⁶. This cross-talk between the two efflux pump operons indicates that in *mtrCDE* overexpression strains where MtrR activity is impaired, the effect of *farA* LOF – like *mtrA* LOF – may be masked.”

6. Past literature has speculated that efflux might be most important in the rectum due to fatty acids and other toxic lipids. Perhaps this should be discussed.

The prior work connecting MtrR with the rectal environment is an important foundation for our findings. We have added discussion of those results in the discussion: **Lines 387-390** “These results complement prior studies suggesting that *mtrR* LOF resulting in increased resistance to fecal lipids plays a critical role in gonococcal adaptation to the rectal environment^{59,60}, and taken together suggest a model in which the fitness benefit of efflux pump expression is highly context dependent.”

7. Supplementary Figure 1, Supplementary Figure 2 legends need more explanations of the figures.

Supplementary Figure 1 caption has been expanded: **Lines 642-650** “**Supplementary Figure 1 – Diagnostic Q-Q plots of expected versus observed p-values for GWAS on a) azithromycin, b) ceftriaxone, and c) ciprofloxacin.** In the absence of confounders such as population structure, p-values are distributed uniformly and would be expected to lie along the y=x line (in red) before diverging at higher $-\log_{10}(\text{p-values})$ due to true causal variants⁹³. Q-Q plots for all three antibiotics appear to be well-behaved, indicating that the steps we have taken to control for population structure (i.e., using a linear mixed model parameterized by the recombination-corrected phylogeny) were adequate. Highly significant markers corresponding to diverging variants at higher $-\log_{10}(\text{p-values})$ were confirmed to map to known causal variants for all three antibiotics (see Supplementary Table 3).”

Supplementary Figure 2 caption has been expanded: **Lines 654-663** “**Supplementary Figure 2 – Alignment of nucleotide sequences for strains with representative LOF mutations observed in a) *mtrC*, b) *mtrA*, and c) *farA* in the global dataset.** The wild-type reference sequences (FA1090 for *mtrC* and *farA*, FA19 for *mtrA*) are shown at the top of the alignment highlighted in yellow. Nucleotide sequences were depicted in black with the corresponding amino acid translations directly under. Dots in LOF sequences represent exact match to the wild-type reference sequence. For *mtrC*, the hexarepeat tract was boxed in the reference genome in green, and mutations leading to LOFs were boxed in red. For *mtrA*, the 11-bp deletion leading to *mtrA* LOF was boxed in red. For *farA*, the repeat tract of Ts was boxed in green, and mutations leading to LOFs were boxed in red. All alignments were visualized in Geneious Prime (see methods).”

Reviewer #2 (Remarks to the Author):

The article by Ma et al makes an important, novel contribution to knowledge regarding antimicrobial

resistance mechanisms in *Neisseria gonorrhoeae* (Ng). The main analysis method was bioinformatics –using a global dataset comprising 4882 genomes to perform genome wide association to identify loss-of-function (LOF) mutations in the MtrCDE efflux pump – i.e. to search for mutations that would cause decreased efflux. The research question in this research is very important - why are some clinical isolates of Ng more susceptible to antibiotics? Quantitative predictions of MICs were based on a well-established method using multiple regression on a panel of resistance markers. The authors identified a loss of function (LOF) mutation in the MtrCDE efflux pump (in *mtrC*) associated with increased susceptibility and the same variant was associated with increased susceptibility to ceftriaxone, ciprofloxacin and other antibiotics. The authors also noted additional LOF mutations associated with increased susceptibility. They also show that the acquisition of antimicrobial susceptibility based on the mutation in *mtrC* was “relatively common” and globally prevalent. Another surprising finding was that the *mtrC* LOF was more prevalent in isolates from men who have sex with women as compared to men who have sex with men, and that this mutation was also more prevalent in cervical isolates. This later finding was validated using a different data set. The authors conclude that environmental pressures in the female reproductive tract may contribute to such a selection. The authors also determined the same associations for isolates with *mtrA* LOF mutations. The authors next hypothesized that environmental acidity could create elective pressure leading to loss of function of the efflux pump. Further analysis with FarAB indicated that mutations in *farA* caused similar associations – i.e. with patient sex of sex partner and cervical isolates. Finally, the authors reported that *mtrC* LOF mutations were enriched in a heterosexual urethritis-associated lineage of *N. meningitidis*.

We thank the reviewer for their comprehensive and thoughtful summary of the key points from our manuscript.

Minor comments:

1. For some data, accession numbers of the data are not provided (e.g. a. *N. meningitidis* ST-11, North America)

Accession numbers and metadata for the ST-11 *N. meningitidis* analysis have now been included as Supplementary Table 8.

2. Perhaps the term “unitig” could be better described.

We have further clarified what this term means in the results: **Lines 108-110** “We used a linear mixed model framework to control for population structure, and we used unitigs constructed from genome assemblies to capture SNPs, indels, and accessory genome elements²⁴⁻²⁶. Unitigs are a flexible representation of the genetic variation across a dataset that are constructed using compacted de Bruijn graphs and have been previously applied as markers for microbial GWAS²⁷.”

3. Parameters for assembly generation (line 421) could be indicated

We specified additional details on the parameters used for assembly generation: **Line 467** “We created *de novo* assemblies using SPAdes (version 3.12.0 run using 8 threads, paired end reads where available, and the --careful flag set)...”

We have also added additional details on the GWAS steps to facilitate reproducibility: **Lines 519-523** “Linear mixed model GWAS was run using Pyseer (version 1.2.0 with default allele frequency filters) with on the 480,902 unitigs generated from GATB (version 1.3.0); the recombination-corrected

phylogeny from Gubbins was used to parameterize the Pyseer population structure random effects term and isolate country of origin was included as a fixed effect covariate.”

4. Line 510 – “supplementary tables” used should be clarified.

We clarified our metadata collection approach: **Lines 568-572** “Patient metadata were collected from the following publications from Supplementary Table 1 that had information on site of infection: Demczuk et al., 2015, Demczuk et al., 2016, Ezewudo et al., 2015, Grad et al., 2014 and 2016, Kwong et al., 2017, Lee et al., 2018, and Mortimer et al., 2020. Sites of infection were standardized across datasets using a common ontology (i.e., specified as urethra, rectum, pharynx, cervix, or other).”

5. Can the authors explain why *farA* and *mtrA* LOFs tended to coincide rather than *farA* and *mtrC* LOFs (Supplementary Figure 3).

We have added discussion on this interesting observation – also shared by Reviewer 1. Please see the response above.

6. It is difficult to read the parts of Figure 2. Would the authors consider adding the regression coefficients for each antibiotic to data in supplementary Table 5? This would be valuable information for investigators who might refer to this research.

We agree that Figure 2 is data rich and potentially confusing and thus have added in additional information to the caption that should make it more interpretable: **Lines 151-163** “**Figure 2 – GWAS identifies a variant mapping to *mtrC* associated with increased susceptibility to azithromycin, ceftriaxone, and ciprofloxacin.** Negative log₁₀-transformed p-values for unitigs tested in GWAS on MICs to azithromycin (pink, n=4505), ceftriaxone (blue, n=4497), and ciprofloxacin (purple, n=4135) are shown in the Manhattan plot. The sign of the GWAS regression coefficient β (with positive indicating an association with increased resistance and negative indicating an association with increased susceptibility) is indicated by symbol shape, as depicted in the legend. Labels indicate known influential resistance determinants, and the *mtrC* variant associated with increased susceptibility was highlighted in gray. A full list of the annotated significant unitigs for each antibiotic can be found in Supplementary Table 3. Inset: schematic of the *mtr* genetic regulon including structural genes *mtrCDE*, the activator *mtrA*, and the repressor *mtrR*. The approximate genomic location within *mtrC* and specific nucleotide change of the *mtrC* GWAS variant relative to the gonococcal NCCP11945 reference genome (i.e., a two base pair deletion in a ‘GC’ dinucleotide repeat) is shown.”

We also appreciate the reviewer’s suggestion to make the data from the GWAS results available. We have included this now as an additional Supplementary Table 3 with variant sequences, regression coefficient and confidence intervals, and annotations for significant variants across all three antibiotics, as well as code on our GitHub to replicate the GWAS results and the corresponding unannotated GWAS outputs.

7. Line 486 – What is Geneious Prime (software available?)

We clarified the source and version for the software Geneious Prime and provided further details for the analysis conducted: **Lines 542-544** “Peptides shorter than 90% of the expected full-length size of

the protein were further analyzed in using Geneious Prime (version 2019.2.1, <https://www.geneious.com>) to identify the nucleotide mutations resulting in predicted LOF by alignment of the nucleotide sequences.”

8. A discussion of the advantages of GWAS over the multiple regression model might be informative for the uninitiated (i.e. perhaps lines 128-131 could be expanded). Perhaps the authors could address machine learning vs GWAS as an analytical tool?

We have added additional information contrasting GWAS with the multiple regression model in the methods: **Lines 512-519** “We used a regression-based GWAS approach to identify novel susceptibility mutations. In particular, we employed a linear mixed model with a random effect to control for the confounding influence of population structure and a fixed effect to control for isolate country of origin. Though the outcome variable (log₂-transformed MICs) is the same, in contrast to the linear modeling approach described above, which models the linear, additive effect of multiple, known resistance mutations, regression in a GWAS is usually run independently and univariately on each variant for all identified variants in the genome, providing a systematic way to identify novel contributors to the outcome variable.” In short, the multiple regression is useful for predictive power and GWAS is useful for identifying new causal markers, though these boundaries are fluid and the methods can be easily combined (e.g., you can also condition on known resistance markers using a conditional GWAS approach). Machine learning would (generally) fall into the former camp with the goal of optimizing predictive power, but this is a broad topic with many available methodologies that is well-addressed elsewhere (see e.g. Hicks et al., 2019 in PLOS Computational Biology) – and hence beyond the scope of our article.

9. The biological relevance of these mutations could be discussed – the numbers of genomes with these mutations was (relatively) small. Would the authors care to comment on whether such mutations make a difference in how women respond to antibiotics?

We thank the reviewer for this thoughtful point. The data connecting sub-threshold MIC changes to clinical outcomes is unfortunately still lacking, but we have expanded our therapeutics application section to reflect this: **Lines 440-446** “Our population-wide estimated effect sizes for *mtrC* LOF mutations provide a prediction for the re-sensitization effect of MtrCDE knockdown across multiple genetic backgrounds and suggest particularly strong effects for the macrolide azithromycin (Supplementary Figure 4). Because the correlation between MIC differences and clinical efficacy is still not well understood, follow up studies to assess treatment efficacy differences in patients with and without *mtrC* LOF strains can help to quantify the expected effect of MtrCDE knockdown in the clinical context.”

Other changes

During the course of manuscript revision, we noticed a small discrepancy in the number of samples included in the linear models and GWAS (n=4882) and the number of samples included in the epidemiological associations (n=4852). This was due to 30 isolates which should have been left out of the GWAS due to failing to meet genomics quality controls. We re-ran the analysis pipeline with the corrected dataset of n=4852 and confirmed that these did not change any of the conclusions in the paper. We have updated the manuscript accordingly with the slight adjustments to prevalence estimates, regression estimates, and p-values.

We have also included analyses from a dataset that was previously embargoed providing additional support for the widespread temporal distribution of *mtrC* LOF mutations: “Because most strains in this dataset were collected within the last two decades, we also examined a dataset of strains collected in Denmark from 1928 to 2013 to understand the historical prevalence of *mtrC* LOF mutations³⁵. We observed an additional 10 strains with the ‘GC’ two base pair deletion ranging in isolation date from 1951-2000, indicating that *mtrC* LOF strains have either repeatedly arisen or persistently circulated for decades.”

REVIEWERS' COMMENTS:

Reviewer #1 (Remarks to the Author):

The authors have responded well to the reviewer's comments and this work is clearly important to the field of *Neisseria* pathogenesis and the real-time evolution of bacterial populations during infection. I intend the following comments to help improve the manuscript, but in my opinion answering these is not essential for publication.

Reviewer 1, Comment 1 asked whether slipped strand mispairing rates were differentially higher in the two sets of data from male and female isolates. In lines 525-528, the new sentence states, "The LOF mutations we observed in *mtrC* and *farA* primarily occurred in short homopolymeric sequences (though with low numbers of repeated units) and thus may occur at a frequency higher than baseline mutation rate, similar to other resistance suppressor mutations⁵⁹." Is there data showing that slippage of a hexanucleotide repeat occurs at a "frequency higher than baseline mutation rate" (note: you shouldn't compare frequency and rate)? I would expect the 8 bp poly-T sequence in *farA* to slip much more frequently, but think that the hexanucleotide repeat and the other shown mutations might be at similar frequencies. The authors only analyzed two identical GC dinucleotide hexarepeats found in two other loci and not overall changes in polynucleotide repeats. I would want to see a comparison of other nucleotide repeat sequences to indicate whether the cervical environment may provide higher rates of slipped strand mispairing or whether this very rare event is under extremely strong selection. I understand that doing this analysis for the entire dataset would be very difficult, but it should be possible to analyze a representative subset of phase-variable, nonessential genes with different type of repeats for a subset of isolates.

The discussion still is missing a clear discussion of what occurs in male and other site isolates and after transmission. The new text discussing mixed infections is good (lines 559-563), and this speculation may suggest that only the efflux competent isolates in a mixed infection transmit to other sites or partners. The alternative hypothesis is that the hexa-nucleotide repeat shifts back to the functional sequence under selective pressure in other sites of colonization. This scenario parallels the idea that strong selection in the cervix is responsible for the loss-of-function mutations. The discussion might consider the possibility that recombination between non-functional and functional alleles occurs in mixed infections.

Reviewer 1, Comment 5: It would be helpful to the reader to provide percentages of overlap between *Mtr* and *Far* mutations.

The new sentence, "The LOF mutations we observed in *mtrC* and *farA* primarily occurred in short homopolymeric sequences (though with low numbers of repeated units) and thus may occur at a frequency higher than baseline mutation rate, similar to other resistance suppressor mutations⁵⁹."

Note: The new Supplementary Figure 2 caption needs italics for gene names.

Hank Seifert

Reviewer #2 (Remarks to the Author):

The authors have very elegantly addressed all reviewer comments with thoughtful and well-reasoned explanations. The paper is much improved because of these modifications and will provide a forum for future experiments and debate about the environmental impacts on the antimicrobial susceptibility of *N. gonorrhoeae* isolates. It is also notable that the authors also noted some discrepancies in their data and have corrected those as well. This highly interesting paper is ready for publication.

Reviewer #3 (Remarks to Author):

Summary

This is a concise, novel and very interesting paper, which combines GWAS with host sample information to conclude that loss of function mutations in an efflux pump lead to adaptation to the female urogenital tract. In short, this is a lovely paper and I support publication immediately.

Details

lines 74/75: my original intention was to comment that while it was very impressive to have collected MIC data spanning 38 years, I was concerned whether MIC was comparable across this timescale - key concentration, inoculum sizes, thresholds might not be consistent across this timescale. I was very impressed to see on line 679 that precisely this issue was addressed.

line 76/77: thanks for doing this step.

line 103-178: section on GWAS. This section is admirably clear and as far as I can see, correctly done. I support the author's choice of using linear mixed models on a uniting graph as a method of handling population structure; Supplementary Figure 1 (which much improved legend) supports their conclusion that the model adequately controls for structure.

line 140 - wonderful that you complemented the mutation and saw MIC increase (stats in Supp table 4 are appropriate)

lines 268-337 - section on association of LOF with cervical environment All of the statistics in this section seem appropriate, and I was very happy to see the independent validation in an Australian cohort, dealing with potential sample bias in the original data line

350-452: sections on activator LOF and proton-dependent efflux pumps Again, all of the statistics look good to me. Clean line of argument and persuasive conclusion at the end of the activator LOF section. On line 430 the authors suggest the phylogenetic distribution of *farA* LOF indicated sustained transmission - this is not clear to me, but I am not very well qualified to comment, and I would defer to other reviewers on this.

lines 456-500: section on meningococcal evolution - again, appropriate statistics applied. line 526 - re rate of homopolymeric slippage - I agree with the authors.

line 698 on - methods section on GWAS Clear and appropriate methods for GWAS. Also good idea to drop to single 23S rRNA copy, and to cluster overlapping unitigs. Delighted to see a GitHub repository with code used, and results, and an iPython notebook.

This is exemplary work, truly exemplary. Can I suggest one small suggestion - in the README, tell the reader to open the iPython notebook in a browser, even if they cannot code in Python, it effectively takes them through the analysis with supporting text and figures.

Conclusion

This has been a rather concise review focussed on the statistics and bioinformatics in the paper as these are my expertise, but the whole paper has been a delight to read - careful and clear work. I fully support publication.

Reviewer comments

Reviewer #1 (Remarks to the Author):

The authors have responded well to the reviewer's comments and this work is clearly important to the field of *Neisseria* pathogenesis and the real-time evolution of bacterial populations during infection. I intend the following comments to help improve the manuscript, but in my opinion answering these is not essential for publication.

We thank the reviewer for the helpful and supportive comments throughout the review process.

Reviewer 1, Comment 1 asked whether slipped strand mispairing rates were differentially higher in the two sets of data from male and female isolates. In lines 525-528, the new sentence states, "The LOF mutations we observed in *mtrC* and *farA* primarily occurred in short homopolymeric sequences (though with low numbers of repeated units) and thus may occur at a frequency higher than baseline mutation rate, similar to other resistance suppressor mutations⁵⁹." Is there data showing that slippage of a hexanucleotide repeat occurs at a "frequency higher than baseline mutation rate" (note: you shouldn't compare frequency and rate)? I would expect the 8 bp poly-T sequence in *farA* to slip much more frequently, but think that the hexanucleotide repeat and the other shown mutations might be at similar frequencies.

We agree that conflating frequencies and rates is incorrect and have removed reference to the former. We have also amended this section to indicate that future experiments will be needed to definitively address whether these shorter repetitive sequences also have higher mutation rates. Prior work in the literature which we referenced (Bichara et al., 2000 in *Genetics*) characterized dinucleotide repeats in *E. coli* and showed that 'GC' dinucleotides were particularly unstable but did not assess repeats as short as a hexarepeat. Hence, our aim here was just to indicate that further investigation of mutation rates in *N. gonorrhoeae* at these sites is warranted. The revised section is now: "The LOF mutations we observed in *mtrC* and *farA* primarily occurred in short homopolymeric sequences (though with low numbers of repeated units) and thus may occur at higher rates than insertions or deletions in non-repetitive regions or nonsense mutations, similar to other resistance suppressor mutations⁴⁷, though this will need to be confirmed in future experiments."

The authors only analyzed two identical GC dinucleotide hexarepeats found in two other loci and not overall changes in polynucleotide repeats. I would want to see a comparison of other nucleotide repeat sequences to indicate whether the cervical environment may provide higher rates of slipped strand mispairing or whether this very rare event is under extremely strong selection. I understand that doing this analysis for the entire dataset would be very difficult, but it should be possible to analyze a representative subset of phase-variable, nonessential genes with different type of repeats for a subset of isolates.

We agree that this type of analysis (a comprehensive comparison of LOF mutation and phase variable sequences across different environmental niches) would be very interesting for the field and are indeed in the middle of conducting it for a future publication. The results from the *mtrC* study certainly point in this direction, but, given the scope, we think it will be better to report on those analyses in a separate study.

The discussion still is missing a clear discussion of what occurs in male and other site isolates and after transmission. The new text discussing mixed infections is good (lines 559-563), and this speculation may suggest that only the efflux competent isolates in a mixed infection transmit to other sites or partners. The alternative hypothesis is that the hexa-nucleotide repeat shifts back to the functional sequence under selective pressure in other sites of colonization. This scenario parallels the idea that strong selection in the cervix is responsible for the loss-of-function mutations. The discussion might consider the possibility that recombination between non-functional and functional alleles occurs in mixed infections.

We agree that further discussion of this would be interesting for readers and have expanded the section on mixed infections to include these hypotheses: “In particular, this intra-host pathogen diversity could facilitate transmission from the female genital tract to other sites of infection, where efflux pump activity incurs less of a fitness cost. In those new sites, isolates with wild-type efflux pump loci in the mixed population could selectively expand relative to LOF efflux pump strains and also serve as possible recombination donors of wild-type alleles. This standing genetic variation would therefore facilitate gonococcal adaptation across different mucosal niches.”

Reviewer 1, Comment 5: It would be helpful to the reader to provide percentages of overlap between Mtr and Far mutations.

We have added additional information on the number / percentage of isolates with LOF mutation overlaps:

“Phylogenetic analysis showed that the distribution of *mtrA* LOF differed from that of *mtrC* LOF with fewer introductions but more sustained transmission and that the two mutations were largely non-overlapping (Supplementary Figure 3); only four strains had both *mtrA* and *mtrC* LOF mutations.”

“The phylogenetic distribution of *farA* LOF indicated sustained transmission (Supplementary Figure 3) and overlapped with that of *mtrA* LOF mutations (48.9% of isolates with *mtrA* LOF mutations also had *farA* LOF mutations), potentially indicating additive contributions to cervical adaptation.”

Note: The new Supplementary Figure 2 caption needs italics for gene names.

Italics have been added to gene names in the Supplementary Figure 2 legend.

Hank Seifert

Reviewer #2 (Remarks to the Author):

The authors have very elegantly addressed all reviewer comments with thoughtful and well-reasoned explanations. The paper is much improved because of these modifications and will provide a forum for future experiments and debate about the environmental impacts on the antimicrobial susceptibility of *N. gonorrhoeae* isolates. It is also notable that the authors also noted some discrepancies in their data and have corrected those as well. This highly interesting paper is ready for publication.

We thank the reviewer for the helpful and supportive comments throughout the review process.

Reviewer #3 (Remarks to the Author):

Can i suggest one small suggestion - in the README, tell the reader to open the iPython notebook in a browser, even if they cannot code in Python, it effectively takes them through the analysis with supporting text and figures.

We thank the reviewer for the thoughtful assessment of the computational and statistical parts of the manuscript. We have updated the README with this helpful suggestion.